# Strategies for Neuroprotection in Multiple Sclerosis and the Role of Calcium

**DOI:** 10.3390/ijms21051663

**Published:** 2020-02-28

**Authors:** Michael Enders, Thorsten Heider, Andreas Ludwig, Stefanie Kuerten

**Affiliations:** 1Institute of Anatomy and Cell Biology, Friedrich-Alexander University Erlangen-Nürnberg, Krankenhausstraße 9, D-91054 Erlangen, Germany; michi.enders@fau.de; 2Department of Neurology, Klinikum St. Marien Amberg, Mariahilfbergweg 7, D-92224 Amberg, Germany; heider.thorsten@klinikum-amberg.de; 3Institute of Experimental and Clinical Pharmacology and Toxicology, Friedrich-Alexander University Erlangen-Nürnberg, Fahrstraße 17, D-91054 Erlangen, Germany; andreas.ludwig@fau.de

**Keywords:** calcium channels, calcium homeostasis, dihydropyridines, multiple sclerosis, neurodegeneration, neuroprotection, nimodipine, remyelination

## Abstract

Calcium ions are vital for maintaining the physiological and biochemical processes inside cells. The central nervous system (CNS) is particularly dependent on calcium homeostasis and its dysregulation has been associated with several neurodegenerative disorders including Parkinson’s disease (PD), Alzheimer’s disease (AD) and Huntington’s disease (HD), as well as with multiple sclerosis (MS). Hence, the modulation of calcium influx into the cells and the targeting of calcium-mediated signaling pathways may present a promising therapeutic approach for these diseases. This review provides an overview on calcium channels in neurons and glial cells. Special emphasis is put on MS, a chronic autoimmune disease of the CNS. While the initial relapsing-remitting stage of MS can be treated effectively with immune modulatory and immunosuppressive drugs, the subsequent progressive stage has remained largely untreatable. Here we summarize several approaches that have been and are currently being tested for their neuroprotective capacities in MS and we discuss which role calcium could play in this regard.

## 1. Role of Calcium in the Central Nervous System (CNS)

### 1.1. Role of Calcium under Physiological Conditions

Calcium is one of the most important second messengers regulating numerous cellular processes such as signal transduction, proliferation, muscle contraction and neurotransmitter release. The calcium concentration differs greatly between the milieu outside (2 mM) and inside (0.00001 mM) the cell [1,2]. 

Perhaps the most prominent example of a calcium-mediated process in the brain is neurotransmitter release. Yet, not only neurons, but also glia cells, which are estimated to be 10 times more frequent than neurons in the CNS, depend on calcium [3]. 

Oligodendrocytes are essential for the function and survival of neurons. Their main purpose is the formation of the myelin sheath [4,5], which is complex and requires different stimuli. The influx of calcium is important for the maturation of oligodendrocyte precursor cells (OPC) and the induction of myelination [6,7,8]. Additionally, the local calcium concentration seems to be essential for myelin elongation and sheath development [9,10,11]. 

Astrocytes are star-shaped glia cells in the brain, which are characterized by their expression of glial fibrillary acidic protein (GFAP). Astrocytes were first thought to be mainly important for providing passive structural support to neurons, but are now well acknowledged to actively participate in the regulation of brain functions [12,13]. They are part of the blood–brain barrier (BBB) [14] and are able to take up sodium and glutamate from the synaptic cleft following the initiation of an action potential [15]. It has been described that an increase in astrocytic intracellular calcium triggers the release of gliotransmitters (neurotransmitters secreted by glia cells) like adenosine triphosphate (ATP), glutamate, D-serine and gamma-aminobutyric acid (GABA) [16]. However, calcium signaling in astrocytes has remained a subject to debate and there are some controversies which have not yet been clarified (reviewed in [16]). 

Microglia are the immune cells of the CNS and play a crucial role after tissue damage by mediating inflammatory processes. An increased extracellular calcium concentration can be recognized by microglia via specific channels, which triggers their migration to the site of the lesion [17]. 

Apart from neurons and glia, many other cells in the CNS depend on calcium. It is clearly beyond the scope of this review to further elaborate on all calcium-mediated cellular processes, but it should be noted that, e.g., the permeability of the BBB, which consists not only of astrocytes but also of endothelial cells and tight junctions, is regulated by intracellular calcium levels [18]. In addition, other immune cells such as T cells, which physiologically populate the human brain, rely on calcium as a second messenger for intracellular processes [19].

### 1.2. Role of Calcium under Pathological Conditions

Dysregulation of calcium homeostasis has been proposed in the pathogenesis of several CNS diseases (reviewed in [20]). One of the general concepts of calcium-induced cell death after damage is excitotoxicity. The term describes induced cell death following an overload of stimulating agents such as glutamate, and the subsequent increase in the intracellular calcium concentration. Excitotoxicity is known to occur in several neurodegenerative diseases including Parkinson’s disease (PD), Alzheimer’s disease (AD) and Huntington’s disease (HD) [20,21,22]. An increased intracellular calcium concentration has been shown to induce superoxide formation and to trigger the release of proapoptotic proteins. It has also been associated with decreased production of ATP and activation of reactive oxygen species (ROS), which can trigger neuronal cell death [23,24]. Additionally, an increased calcium concentration activates calmodulin-dependent kinases and the production of nitric oxide (NO) [23,24,25,26].

## 2. Calcium Channels Expressed by the Different Cell Types of the CNS

### 2.1. Intracellular Calcium Channels

Intracellularly, there are three major groups of channels which are responsible for the release of calcium from the endoplasmic reticulum (ER): ryanodine receptors (RyR) [2], inositol-3-phosphate receptors (IP3R) [2,27] and two-pore channels (TPC) [28]. 

#### 2.1.1. RyR

RyR are named after their most prominent binding partner, ryanodine. RyR activity is mainly triggered by an increased intracellular calcium concentration originating from the extracellular side [1]. The three different isotypes of RyR (RyR1-3) are all expressed in the brain. While RyR3 is widely distributed inside the brain [29], RyR2 seems to be the most abundant receptor type [30,31]. All RyR are present in neurons, but their distribution is different regarding cellular localization, brain area and stage of development [32]. For example, in granule cells of the cerebellum only RyR2 has been found [33], whereas Purkinje cells express all RyR subtypes [34]. In OPC, only RyR3 is present, which is then downregulated during development [35]. In mature oligodendrocytes, all RyR have been reported to be present [36]. The pattern of RyR expression in astrocytes is unclear. One study found that only RyR3 is expressed in astrocytes [37]. In contrast, another study has observed all RyR subtypes in rat brain astrocytes [35]. Microglia are known to express functional RyR1 and RyR2 channels, while RyR3 is only detectable in fetal stages of microglia during brain development [38].

#### 2.1.2. IP3R

IP3R are named after their agonist inositol trisphosphate (IP3). IP3R activation is mainly mediated by G-protein coupled receptors (GPCR) [1]. Three different isotypes of IP3R exist: IP3R1, IP3R2 and IP3R3. Every brain cell type contains at least one type of IP3R [2]. Most of the neurons in the CNS contain IP3R1 [39], which is expressed throughout the cell, and IP3R3, which is expressed only in the soma [40,41]. IP3R2 is exclusively located in glia cells [41], especially in astrocytes [42]. 

#### 2.1.3. TPC

TPC are nonselective cation channels conducting sodium and calcium [43] They are mainly located in the membrane of acidic vesicles [44], where they mediate calcium homeostasis after activation by nicotinic acid adenine dinucleotide phosphate (NAADP) [28]. In neuronal cells, the channel has been described to mediate differentiation [45] and it also seems to be important for autophagic processes in astrocytes [46,47]. 

### 2.2. Plasma Membrane Calcium Channels 

There are several different calcium channels within the plasma membrane which are important for the function of CNS cells, including voltage-gated calcium channels (VGCC) [48], ionotropic glutamate receptors [49,50], transient receptor potential (TRP) channels [51], calcium release-activated calcium (CRAC) channels [52] and purinergic P2X receptors [53]. 

#### 2.2.1. VGCC

VGCC comprise a range of different channels: L-type (Ca_V_1.1–Ca_V_1.4), P/Q-type (Ca_V_2.1), N-type (Ca_V_2.2), R-type (Ca_V_2.3) and T-type (Ca_V_3.1–Ca_V_3.3) channels. The biophysical properties of each class are determined by the pore-forming α-subunit and associated auxiliary subunits (reviewed e.g., in [48,54]). All VGCC are activated by depolarization of the membrane potential. VGCC are widely distributed in neurons and glia cells [48]. 

Ca_V_1.2 and Ca_V_1.3 are the most abundant L-type channels in the brain. They are located at postsynaptic dendrites and the soma [55] and control neuronal excitability and gene transcription [56]. Ca_V_2.1 and Ca_V_2.2 are located at the presynapse, where they mediate transmitter release [57], whereas Ca_V_2.3 is mainly located on neuronal spines [58]. T-type channels are expressed on the cell body and control the rhythmic burst firing of thalamic neurons, which is important for the generation of neuronal oscillations under physiological (sleep states) and pathophysiological (absence epilepsy) conditions [48,54]. 

The expression profile of VGCC changes during development from OPC to mature oligodendrocytes [59,60]. VGCC are expressed on OPC and mature oligodendrocytes, while being absent on immature oligodendrocytes [61]. Whereas low VGCC (probably T-type) are present at OPC extensions, high VGCC (probably L-type) are mostly located at the soma of OPC [60]. In mice, it has not only been shown that after knockout of Ca_V_1.2 less myelination of axons occurred and the maturation from OPC to oligodendrocytes was disturbed [7], but also remyelination was impaired [62]. Remarkably, the expression of myelin basic protein (MBP), an essential myelin protein, has been demonstrated to inhibit the influx of calcium via VGCC, whereas the so-called golli forms (precursors of MBP, which are present in OPC) enhanced the influx of calcium via VGCC [63,64]. Additionally, the N-type Ca_V_2.2 was detected on mature oligodendrocytes [65].

Furthermore, astrocytes express VGCC [66]. One of their functions in astrocytes is the release of gliotransmitters like glutamate [67]. However, there are several studies which provide contradictory results regarding the expression profile of VGCC on astrocytes. All studies have shown the expression of Ca_V_1.2. In addition, Cheli et al. have demonstrated the expression of Ca_V_2.1 and Ca_V_1.3 on mouse primary astrocytes [68]. Another study was unable to detect Ca_V_2.1 on rat primary astrocytes but observed the expression of Ca_V_1.3, and of Ca_V_2.2, Ca_V_2.3 and Ca_V_3.1 [69]. However, D’Ascenzo et al. did not detect Ca_V_1.3, but Ca_V_2.1, Ca_V_2.2 and Ca_V_2.3 on primary astrocytes from rats [70]. Taken together, it seems reasonable that the expression of VGCC on astrocytes is species-specific and dynamic, brain region- and/or development-dependent. 

Microglia are predicted to express at least L-type VGCC, but the exact isoform has not been identified so far [71].

#### 2.2.2. Ionotropic Glutamate Receptors

The three ionotropic glutamate receptors, α-amino-3-hydroxy-5-methyl-4-isoxazolepropionic acid (AMPA), N-methyl-d-aspartate (NMDA) and kainate (KA) receptors [72], are essential for synaptic transmission and plasticity [73,74,75]. For a more detailed overview on their molecular structure, their subcellular compartment and their physiological properties, please refer to comprehensive reviews on AMPA [76], NMDA [77] and KA [78] receptors. All three receptors have been detected on oligodendrocytes [79,80], and excitotoxicity in these cells seems to be mediated by AMPA and KA receptors [81,82]. Astrocytes have AMPA and KA, but no NMDA receptors [83,84]. Microglia express AMPA and KA receptors [85] and at least a subunit of NMDA receptors with unknown function [86]. 

#### 2.2.3. TRP Channels

TRP channels have a broad distribution profile and are non-selective cation channels, with some of them being permeable for calcium. The superfamily of TRP channels consists of 28 members in humans, which are organized in six subgroups: TRPA, with A for “ankyrin”; TRPC, with C for “canonical”, TRPM, with M for “melastatin”, TRPML, with ML for “mucolipidin”, TRPP, with P for “polycystin” and TRPV, with V for “vanilloid” [87]. 

The TRPA class consists of only one receptor type: TRPA1. TRPA1 is expressed in all cortical layers as well as in the hippocampus of rodents, and it has been shown that TRPA1 can affect neuronal circuits [88]. Additionally, there are reports on TRPA1 expression on oligodendrocytes [89] and astrocytes [90]. 

Channels of the TRPC family which are known to be expressed in the brain are TRPC1, TRPC4, TRPC5 [91], TRPC3 and TRPC6 [92]. While TRPC1, TRPC4 and TRPC5 are activated via IP3, TRPC2, TRPC3, TRPC6 and TRPC7 are activated via diacylglycerol (DAG) [93]. TRPC1 expression has been described on OPC, where it mediates OPC differentiation [94]. TRPC3 is located on oligodendrocytes [95]. TRPC1 [96], TRPC3 [97] and TRPC6 [98] are present on astrocytes, while TRPC3, especially, seems to be involved in gliosis after neuronal damage [97]. Microglia are known to increase their expression of TRPC3 after brain-derived neurotrophic factor (BDNF) stimulation [99] and TRPC6 activation by amyloid β (Aβ) has been reported [100]. 

TRPM1 is found in very small quantities in the CNS and its function has remained unknown [92]. TRPM2 channels are expressed on neurons. Here, they mediate synaptic plasticity as well as oxidative stress and their expression is increased after interleukin stimulation [101]. TRPM3 and TRPM5 are expressed on Purkinje cells, whereby the former is predicted to play a role in early brain development [102,103]. TRPM7 is also highly expressed in the brain [92], where it plays an important role in learning and memory processes in hippocampal neurons [104]. Oligodendrocytes or at least OPC are known to express TRPM3, which can be actively involved in myelination [105]. TRPM2, TRPM4 and TRPM6 are present on astrocytes [106]. TRPM2 expression is increased after interleukin stimulation [101]. Reactive astrocytes express high levels of TRPM7, which seems to be important for gliotic scar formation [107]. Like TRPC3, TRPM1 is upregulated after stimulation of microglia with Aβ [108], whereas TRPM2 is involved in microglia activation [109] after interleukin stimulation [101]. TRPM7 is important for migration properties [110]. Additionally, the TRPM4 channel was found on microglia, but its function remains unclear [111]. 

TRPV channels mediate pain transduction in nociceptors. All TRPV channels (TRPV1 to TRPV6) are expressed in the mouse brain [112]. While TRPV1 to TRPV4 are expressed on neurons, TRPV5 and TRPV6 are mainly found in epithelial tissue [113]. For OPC, only the expression of TRPV4 has been reported, and channel activity was shown to increase OPC proliferation [114]. Mature oligodendrocytes also express TRPV1 [115]. Astrocytes express four types of TRPV channels: TRPV1, which is involved in astrogliosis [116], TRPV2, which is important for the regulation of the lipid metabolism [117], TRPV4, which is osmosensitive and regulates brain homoeostasis [118] and TRPV6 with unknown function [106]. Microglia or at least the BV2 microglia cell line are known to express TRPV1 and TRPV2, which mediate phagocytosis [119]. TRPV4 is additionally involved in microglia activation [120]. 

#### 2.2.4. Calcium Release-Activated Calcium Channels

The family of CRAC channels consists of three different members: Orai1, Orai2 and Orai3 [52]. The influx of extracellular calcium into the cytosol as well as the release of internal calcium from the ER mediated by Orai channels drives exocytosis, stimulates mitochondrial metabolism, activates gene expression, and promotes proliferation [121]. The stromal interaction molecule 1 (STIM1) is important for the function of Orai1. It is located in the membrane of the ER and acts as a calcium sensor for increased calcium concentration in the cytoplasm. STIM1 interacts with Orai1, which is indispensable for the pore opening of the Orai1 channel [122]. This process is termed store-operated calcium entrance (SOCE). SOCE occurs in neurons upon Orai1 and Orai2 interaction with STIM1 and/or STIM2 [123], and this process appears to be essential for neuronal growth [124]. Oligodendrocytes and astrocytes (at least in the optic nerve of mice) also express Orai1 as well as STIM1 and STIM2 [125]. Microglia express Orai1 and STIM1 [126]. For a detailed review on Orai and STIM function see [127].

#### 2.2.5. Purinergic P2X Channels

P2X receptors belong to the family of purinergic receptors. They are ionotropic and activated by ATP [128,129]. P2X channels are located at presynaptic and postsynaptic nerve terminals [129]. Regarding the different subtypes of P2X channels in neuronal cells see [53,128]. A detailed summary of P2X receptors is provided in [130]. A functional analysis of purinergic receptors in oligodendrocytes and microglia is provided in [131] and [132], respectively.

Figure 1 summarizes the expression profile of calcium channels by the different cell types of the CNS.

## 3. Pharmacological Blockers of Calcium Channels in the CNS

### 3.1. Dihydropyridines (DHP) 

In 1969, the German physiologist Albrecht Fleckenstein described the effect of Bay a 1040 on calcium influx in the heart as “Ca^++^-antagonism”, later known to be due to a specific block of L-type VGCC [133]. This compound was named nifedipine and is the first in the drug class of DHP. Further drugs in this series included amlodipine, nicardipine, nimodipine and lercanidipine [134]. DHP block all L-type calcium channels (Ca_V_ 1.1–Ca_V_ 1.4), whereas P-, N-, and R-type (Ca_V_ 2.1–Ca_V_ 2.3) channels are relatively unaffected [48]. Some DHP, including nifedipine, amlodipine and lercanidipine, are important anti-hypertensive drugs [135], while other DHP like isradipine or nimodipine may have a beneficial effect in CNS-related diseases [48]. A proposed general effect of DHP action in CNS disorders could be the stimulation of autophagic processes and the clearance of toxic components like mutated huntingtin or tau protein [136]. A recent review summarizes the potential effects for each DHP and for each channel in detail in the context of CNS diseases including AD, HD, PD and psychotic disorders [48]. 

### 3.2. Other Calcium Channel Blockers Modulators Affecting the CNS 

Besides DHP, other drugs act on L-type VGCC. Phenylalkylamines like verapamil, which have a very limited BBB permeability, and benzothiazepines like diltiazem, block L-type calcium channels [137]. In addition, several antiepileptic drugs including ethosuximide, valproic acid and zonisamide, and even some antipsychotic drugs like pimozide and penfluridol block T-type calcium channels, which may explain at least parts of their clinical effects [138]. Gabapentine and pregabalin, two anticonvulsant drugs which are also used for neuropathic pain treatment, block VGCC via binding to the α2δ subunit [139]. Not only chemically synthesized small molecules interact with VGCC. A highly specific blocking action has been found for various peptide toxins including ω-conotoxin and ω-agatoxins. A list with some toxins which bind to VGCC channels can be found in [140]. For other pharmacologically active substances that modulate calcium homeostasis, please refer to reviews on TRP channels [141,142], purinergic receptors [143], IP3R and RyR [144]. 

## 4. General Overview of Multiple Sclerosis (MS)

### 4.1. Clinical Course of MS

MS is the most common neurological CNS disease in young adults [145]. Due to irreversible deficits, premature retirement, long-term medication and nursing, MS causes a high socioeconomic burden [146]. The disease is characterized by an inflammatory response against oligodendrocytes, which eventually leads to demyelination and axonal damage and loss [147,148]. Around 85% of all patients initially show a relapsing-remitting course of MS (RRMS). Furthermore, 15% of MS patients suffer from the primary progressive disease subtype (PPMS), which is characterized by constant worsening of clinical symptoms [147]. Most patients with RRMS transit into a secondary progressive form of MS (SPMS) after 8 to 20 years [147]. While RRMS is treatable using anti-inflammatory and immune modulatory drugs, treatment options for SPMS are currently limited to the drug siponimod, which has recently been shown to have neuroprotective capacity in a phase 3 clinical trial [149].

### 4.2. Etiology of MS

The cause of MS is unknown. Yet, there are several theories regarding the factors that could trigger MS. These include (1) genetic predisposition and single nucleotide polymorphisms in genes which are involved in immune function [150], (2) viral infections, in particular with Epstein-Barr virus or bacteria including *Helicobacter pylori* [151], (3) vitamin D deficiency [152] as well as (4) a dysbalanced microbiome and pathology of the enteric nervous system [153]. 

### 4.3. Pathophysiology of MS

The pathophysiology of MS differs in patients with early stage RRMS compared to SPMS [154]. In RRMS, T cells, in particular of the T_H_1 and T_H_17 type, are assumed to be involved in early disease development [154]. CD4^+^ cells are presumably activated in the periphery before they cross the BBB to initiate the immune response in the CNS. After breakdown of the BBB, other immune cells like CD8^+^ T cells, B cells and macrophages are attracted, causing edema and a diversification of the immune response [155]. In later stages of the disease, meningeal B cell aggregates may locally contribute to the immunopathology [156]. In SPMS, infiltration from the periphery gradually wanes and neurodegenerative processes inside the brain prevail. The absence of clinical, imaging, immunological and clear-cut pathological criteria that define the transition from relapsing-remitting to progressive disease explains why SPMS can only be diagnosed retrospectively in most of the cases [157] and why there is still a lack of treatment strategies for late-stage MS [158].

### 4.4. Treatment of MS

While MS has remained incurable, RRMS has become treatable [158]. The main goal in the treatment of RRMS is anti-inflammation, immune modulation and the inhibition of immune cell infiltration into the CNS. This is achieved by disease-modifying therapies (DMT). The use of DMT is not uniformly regulated and guidelines for MS treatment can differ between countries. In the following, the drugs which are most commonly used in Europe are listed in alphabetical order: alemtuzumab, a monoclonal anti-CD52 antibody [159]; cladribine, which depletes both B and T cells; dimethyl fumarate, which is proposed to inhibit nuclear factor kappa-light-chain-enhancer of activated B cells (NF-κB) signaling and to alter immune cell activation; fingolimod, a sphingosin-1-phosphate receptor (S1PR) agonist which inhibits immune cell emigration from secondary lymphoid organs [160]; glatiramer acetate [161] and interferon-β [162], two immune modulatory drugs; natalizumab, a potent α_4_-integrin inhibitor which prevents lymphocyte migration over the BBB [160]; mitoxantrone, a cytostatic agent [163]; ocrelizumab, a monoclonal anti-CD20 antibody which depletes B cells; and teriflunomide, which has anti-proliferative action on immune cells [160]. 

## 5. The Role of Calcium in MS

The focus of research and treatment in MS has been on the reduction of immune cell infiltration into the CNS for a long time, leading to the discovery and development of several DMT. Yet, there is urgent need for the development of neuroprotective and neuroreparative strategies to prevent long-term disease progression and disability, and with that the socioeconomic burden. 

Several treatment strategies along these lines have already been and are currently being tested in MS and other neurodegenerative diseases. While in many instances promising results were obtained in preclinical studies using cell culture and animal models, so far only the drug siponimod has made its way into clinical application for treatment of SPMS patients with its recent approval in the US and in Europe [147].

In the following we would like to review and discuss why calcium may be a reasonable therapeutic target when it comes to neuroprotection in MS. 

### 5.1. Calcium and Excitotoxicity

As described above, excitotoxicity is known to occur under pathological conditions and has been associated with both experimental autoimmune encephalomyelitis (EAE) and MS [164]. AMPA and KA receptors are involved in the pathological pathway of excitotoxicity, and an inhibition of these receptors can decrease EAE severity [24]. One reason for increased glutamate receptor activity in EAE could be that T cells either interfere directly with the receptors or enhance glutamate transmission through the release of tumor necrosis factor α (TNF-α) [165]. TNF-α also activates glutamate release from microglia which can further trigger excitotoxicity [166]. Another concept of excitotoxicity suggests altered glutamate uptake by glia cells. Astrocytes have shown decreased glutamate uptake in the presence of interleukin 17, a signature cytokine in EAE and MS, and increased calcium-dependent release of glutamate [167]. In another study, disturbed glutamate uptake in oligodendrocytes, which can be induced by chemicals, could be prevented by AMPA antagonists [168]. In post-mortem brains of MS patients an accumulation of activated microglia was observed in areas in which glutamate uptake was disturbed [169]. Elevated levels of glutamate can, in turn, trigger an increase in the intracellular calcium concentration, which again causes excitotoxicity by ROS production and calcium overload in mitochondria [170]. Because the TRPM2 channel is very sensitive to oxidative stress and activation of microglia is mediated by this channel, targeting TRPM2 might reduce excitotoxic processes [109]. 

### 5.2. Calcium and BBB Integrity

The BBB consists anatomically of smooth muscle cells, endothelial cells and astrocytes, forming a natural barrier that protects the brain. It also plays a role in regulating calcium homeostasis. The tight and adherent junctions, which are present between the endothelial cells of the BBB as well as the actin cytoskeleton inside the cells, are dependent on the intracellular and extracellular calcium concentration [171,172]. After damage, the intracellular calcium concentration increases due to GPCR and IP3 signaling in endothelial cells, subsequent activation of SOCE and a direct influx of calcium via calcium channels. This leads to the reorganization and/or modification of the actin cytoskeleton and tight junctions with subsequent BBB leakage [18,173]. Interestingly, cannabinoid CB2 receptor agonists have been shown to enhance the formation of tight junctions and to reduce leakiness of the BBB [174]. In addition, activation of the calcium/calmodulin-dependent protein kinase (CaMK) has been reported to promote BBB repair after stroke in a mouse model, which might prevent immune cell infiltration [175]. 

### 5.3. Calcium and Immune Cell Activation

Calcium homeostasis in T cells is important for T cell action. The T cell receptor can trigger intracellular calcium release via SOCE. TRP channels act as regulators on T cells, VDCC channels modulate their activity and purinergic receptors amplify the T cell response [176]. The complex interplay of calcium channels on and in T cells is reviewed in [176]. Depletion of STIM1 and STIM2 has been shown to protect mice from EAE [177]. The same was observed after knockout of the Ca_V_3.1 channel in T cells, which was accompanied by altered cytokine secretion and reduced gene transcription [178]. Furthermore, the effect of vitamin D is currently under investigation in MS patients who frequently display decreased vitamin D levels [179]. Vitamin D supplementation seems to reduce neuronal damage after oxidative stress in cell culture [180] and has also been investigated regarding its impact on immune cells [181]. While vitamin D supplementation and subsequent hypercalcemia was counterproductive in EAE mice, a lower concentration seemed to be slightly beneficial [182]. 

### 5.4. Drugs that Directly Interfere with Calcium Homeostasis and May Have Neuroprotective Properties in MS

#### 5.4.1. Olesoxime

The drug olesoxime (TRO19622) has the potential to interfere with calcium homeostasis by binding to pore proteins on the outer mitochondrial membrane [183]. The drug has revealed a remyelinating effect in several demyelinating rodent models [184,185] and also in a mouse model of amyotrophic lateral sclerosis (ALS) [186]. However, a clinical trial in ALS patients failed [187]. In a phase 1 study for spinal muscle atrophy (SMA) the drug was well tolerated [188], and a long-term study was completed (NCT02628743), from which results are estimated soon. A MS study was additionally performed (NCT01808885), but so far results have not been published. 

#### 5.4.2. Quetiapine

Quetiapine is an antipsychotic agent, which is used for the treatment of schizophrenia and bipolar disorders. Quetiapine has been shown to attenuate EAE severity in mice due to a reduction in the number of T cells in peripheral lymphoid organs [189] and a positive effect on oligodendrocyte development and myelin regeneration [190,191,192]. Quetiapine is suggested to have an impact on calcium homeostasis by interacting with STIM1 [193] and to improve remyelination through this pathway [193,194]. Clinical phase 1 and 2 trials in MS patients are ongoing (NCT02087631).

#### 5.4.3. DHP 

The L-type VGCC antagonist nimodipine is an interesting candidate for neuroprotection in MS and effects on CNS and PNS have been reported early on [195]. As described above, calcium influx via VGCC is important for neuronal excitability, gene expression, regulation of transcription factors and post-transcriptional modifications. Some subunits of VGCC can even act as transcription factors themselves and several so-called “channelopathies“ have been connected to mutations in VGCC genes [196].

Nimodipine is approved for the prevention of vasospasms in patients with subarachnoid hemorrhage. It can cross the BBB and enter the CNS [197]. A recent study has demonstrated a direct correlation between the concentration of nimodipine in cerebrospinal fluid, but not in arterial blood, and the disease outcome after subarachnoid hemorrhage [198]. These data may indicate a protective effect of nimodipine. 

Further studies on the effects of nimodipine in the CNS have been performed in animal models, demonstrating an improvement of working memory not only in old [199], but also in young rats [200]. In addition, cognitive dysfunction after surgery could be prevented by treating rats with nimodipine before surgery [201]. Nimodipine also increased cognitive function after cerebral ischemia in different animal models [202] and displayed a neuroprotective effect on dopaminergic neurons in mouse and rat cell cultures [197] indicating a potential usefulness for the treatment of PD [203]. Finally, nimodipine has been shown to induce remyelination in mouse models of MS [204,205]. This remyelinative effect was connected to microglia-specific apoptosis and diminished production of NO and ROS [204]. When nimodipine was administered preventively at the time point of disease induction, EAE severity was decreased and the extent of demyelination decreased [205]. In yet another study, a neuroprotective effect of nimodipine treatment was observed when the drug was prophylactically given before vestibular schwannoma surgery [206]. 

Recently, nimodipine has been reported to have positive effects on Schwann cells, astrocytes and neurons, which was associated with increased phosphorylation of proteinkinase B and the cyclic adenosine monophosphate response element-binding protein (CREB) [207,208,209]. Additionally, the pro-apoptotic protein caspase 3 and the calcium-dependent protein calpain were downregulated upon nimodipine treatment, whereas calbindin expression was upregulated, indicating that modulation of calcium homeostasis and prevention of calcium overload might be responsible for the neuroprotective properties of nimodipine [207,210]. 

Nimodipine may also act independently of VGCC and calcium, as suggested by effects on microglia, which do not possess Ca_V_1.2 [204]. Along these lines, it has been suggested that nimodipine may inhibit microglia activation through an effect on the purinergic P2X_7_ receptor [211]. 

### 5.5. Potential Neuroprotective Drugs that Indirectly Interfere with Calcium Homeostasis via GPCR

GPCR are the most important family of receptors that activate calcium signaling upon external stimuli, making them ideal drug targets. In the following, several GPCR that could be modulated towards neuroprotection in MS are discussed along with potential drug candidates.

#### 5.5.1. Interference with Cannabinoid CB1 and CB2 Receptors

Cannabidiol, which is an agonist at cannabinoid CB1 and CB2 receptors, has been shown to protect proteins from oxidative stress and to increase mitochondrial activity (reviewed in [212]). Moreover, the synthetic cannabinoids WIN 55,212-2, arachidonyl-2-chloroethylamide (ACEA), and JWH-015 have been reported to decrease inflammatory responses and to increase remyelination in the Theiler’s murine encephalomyelitis virus model [213]. WIN 55,212-2, a cannabinoid CB1 receptor agonist, has additionally been demonstrated to have a protective effect on granule cells of the dentate gyrus and it also interferes directly with Ca_V_2.2 and TRPA1 at high concentrations [214]. The effect of ACEA is potentially mediated by inhibition of the mitochondrial permeability transition pore via cannabinoid CB1 receptor signaling [215]. For JWH-015, a cannabinoid CB2 receptor agonist, a similar mode of action is assumed [216], and an effect on potentially neuroprotective genes has been described [217]. The synthetic compound Yhhu4952 has been demonstrated to interact with OPC possibly via the cannabinoid CB2 receptor, enhancing their maturation [218]. 

#### 5.5.2. Antagonism of Histamine Receptors

Clemastine, a histamine H1 receptor antagonist, has been investigated for remyelination in mice [219]. Additionally, the ReBUILD study has investigated the drug in optic neuritis [220]. The ReCOVER study is currently ongoing (NCT02521311). In addition, the two histamine H3 antagonists GSK239512 [221] and GSK247246 [222] were investigated for their remyelinating properties in mice. 

#### 5.5.3. Stimulation of the κ Opioid Receptor (KOR)

Activation of KOR has been connected to an inhibition of N-type calcium channels [223]. Moreover, the KOR agonist U-50488 directly blocks several low and high VGCC, however concentrations needed for this effect are much higher (100×) than for KOR activation [224]. 

Thus, KOR agonists are a potentially interesting target for controlling calcium homeostasis. Asimadoline, as well as U-50488, has been shown to induce remyelination in an EAE mouse model via KOR activation [225], however the suggested mechanism involved the MAP kinase pathway rather than a modulation of calcium channels [226]. 

#### 5.5.4. Antagonism of Muscarinic Receptors 

Benzatropine is a muscarinic M1/M3 receptor antagonist, which induces remyelination in EAE models [227,228]. Interestingly, the drug has shown no effect on the immune system including T cells, which suggests that the remyelinating properties of the drug are not due to immune modulation [228].

#### 5.5.5. Vitamin D

Supplementation of vitamin D has been shown to have a beneficial impact on EAE outcome in mice [229]. This effect was attributed to an increase of anti-inflammatory and a decrease of pro-inflammatory cytokines [229]. In addition, vitamin D was reported to confer neuroprotection in parallel with downregulation of L-type VGCC in hippocampal neurons [230]. The potential benefits of vitamin D supplementation in MS patients are still under investigation [152].

The following Table 1 and Table 2 summarize some key information on the drugs which may be neuroprotective in MS due to direct or indirect effects on calcium.

## 6. Conclusions

In this review we aimed to illustrate the importance of calcium homeostasis in the CNS. This importance is not only reflected by numerous cellular processes which depend on calcium, but also by the sheer amount of calcium-associated channels and receptors, which are expressed by various CNS cell types. Dysregulation of calcium homeostasis is a pathogenic feature of several neurodegenerative diseases making it an interesting target for novel treatment approaches. MS is the most prevalent neurological disease of the CNS in young adults and there is hope that drugs interfering with calcium homeostasis may be a valuable option for the progressive disease, which is still untreatable, but causes high socioeconomic burden.

First data, in particular regarding desirable effects of DHP on brain pathology, exist. DHP encompass a well-characterized family of drugs, with several members showing BBB permeability. Future research will have to focus on the diseases which may benefit most from DHP treatment, going hand-in-hand with an in-depth characterization of drug-mediated effects and side effects. Nimodipine has shown neuroprotective effects in the EAE model [204] and in patients with vestibular schwannoma [206]. Since the drug has already been on the market for a long time and its safety profile is well established, a clinical trial in progressive MS seems feasible in the near future.

Beside DHP, the two drugs clemastine and quetiapine are currently being tested in SPMS patients. In addition, a trial on benzatropine could be worth considering since a remyelinating effect has also been observed in EAE.

## Figures and Tables

**Figure 1 ijms-21-01663-f001:**
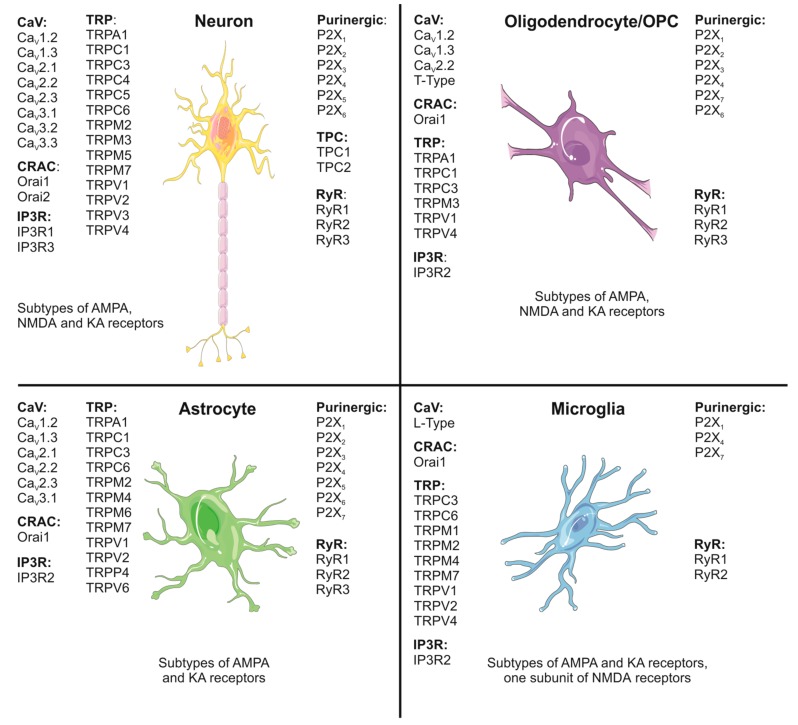
Expression profile of calcium channels in mammalian central nervous system (CNS) cell types. The image shows neurons (yellow), oligodendrocytes (purple), astrocytes (green) and microglia (blue). For a detailed description please refer to the text.

**Table 1 ijms-21-01663-t001:** Drugs which directly interfere with calcium signaling and may potentially be neuroprotective in multiple sclerosis (MS).

Drug	Mode of Action	Proposed Effects	Disease (Model)	Species	NCT	Obvious Drawbacks	References
Nimodipine	Dihydropyridine blocking L-type voltage-gated calcium channels (VGCC)	Increased oligodendrocyte precursor cell (OPC) numbers, microglia-specific apoptosis, reduction of nitric oxide (NO) and reactive oxygen species (ROS) production, increased remyelination	Experimental autoimmune encephalomyelitis (EAE)	Mouse	-	-	[204,205]
Olesoxime	Cholesterol targeting voltage-dependent anion channels (VDAC) in the outer mitochondrial membrane	Neuroprotective agent affecting cytosolic calcium homoeostasis	EAE	Mouse, rat	NCT02628743 NCT01808885	No long-term experience, not approved for any disease	[183,184,185,186,187,188]
Quetiapine	Antagonist at multiple G-protein coupled receptors (GPCR) (e.g., histamine H1, dopamine D2, 5-HT2A), interaction with mitochondrial calcium channel STIM1	Increased maturation of oligodendrocytes	Cuprizone model, MS	Mouse, human	NCT02087631	-	[190,191,192,193,194]

**Table 2 ijms-21-01663-t002:** Drugs which indirectly interfere with calcium signaling and may potentially be neuroprotective in MS.

Drug	Mode of Action	Proposed Effects	Disease (Model)	Species	NCT	Drawback	References
Arachidonyl-2-chloroethylamide (ACEA)	Cannabinoid CB1 receptor agonist	Inhibition of the mitochondrial permeability transition pore leading to neuroprotection by decreased calcium influx into the cytosol	Theiler’s murine encephalomyelitis virus model	Mouse	-	No long-term experience	[213,215]
Asimadoline	κ Opioid Receptor (KOR) agonist	Induction of remyelination	EAE	Mouse	-	Low blood–brain barrier (BBB) permeability	[225]
Benzatropine	Muscarinic M1/M3 receptor antagonist	Enhanced remyelination via the induction of OPC differentiation	Cuprizone model, EAE	Mouse	-	-	[227,228,231]
Clemastine	Histamin H1 receptor antagonist	Induction of remyelination	Cuprizone model, patients with acute optic neuritis	Mouse, human	NCT02521311	-	[219,220,232,233]
GSK239512	Histamine H3 receptor antagonist	Small but positive effect on remyelination in a phase 2 study	Relapsing-remitting MS (RRMS)	Human	NCT01772199	No long-term experience	[221]
GSK247246	Histamine H3 receptor antagonist	Induction of remyelination	Cuprizone model	Mouse	-	No long-term experience	[222,234]
JWH-015	Cannabinoid CB2 receptor agonist	Inhibition of the mitochondrial permeability transition pore leading to neuroprotection by decreased calcium influx into the cytosol	Theiler’s murine encephalomyelitis virus model	Mouse	-	Potentially psychoactive, illegal in some countries, no long-term experience	[213,216,217]
U-50488	KOR agonist	Increased remyelination	EAE	Mouse	-	No long-term experience	[225,233,235,236]
Vitamin D	Vitamin D receptor stimulation	Unclear neuroprotective mechanism, evidence for the downregulation of L-type VGCC	EAE, hippocampal neurons, MS	Mouse, rat, human	-	Uncertain effects	[229,230,237]
WIN 55,212-2	Cannabinoid CB1 receptor agonist	Potential neuroprotective effects by modulation of TRPA1 and Ca_V_2.2 activity	Theiler’s murine encephalomyelitis virus model	Mouse	-	Potentially psychoactive, illegal in some countries, no long-term experience	[213,214]
Yhhu4952	Cannabinoid CB2 receptor agonist	Improved remyelination, increased OPC maturation in culture by alteration of the Notch1 pathway, reduction of BBB leakiness	Cuprizone model	Neonatal rats, mouse	-	No long-term experience	[218,238]

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
