# Peer review of "Strategies for Neuroprotection in Multiple Sclerosis and the Role of Calcium"

_ijms, 2020, doi:10.3390/ijms21051663_

Round 1
Reviewer 1 Report
This is a very valuable review. The introduction to the calcium channel receptors was very helpful, and the extensive acknowledgement of other reviews was welcome.
In the section on MS
page 7 - line 260 "wanes" not "vanes"
section 4.3- the distinction between RRMS and SPMS needs to be softened. Modern thinking rejects the idea of a sudden switch from relapsing pathology to progressive pathology. This could be rewritten. I suggest citing the Lublin paper
Lublin, F.D., et al., Defining the clinical course of multiple sclerosis The 2013 revisions. Neurology, 2014. 83(3): p. 278-286.
4.4 The discussion re second line treatments reflects a local view that does not have world wide applicability- suggest remove the concept of "second line" and think of some other term. ie not all countries have the same restrictions on prescribing
5.2 This section talks about calcium channels on endothelium and immune cells. This needs to be introduced in the section about channels on brain cells
5.5.2 and 5.5.4- I cannot see the connection between clemastine and muscarinic receptors and calcium channels
Author Response
Reviewer #1
We would like to thank the Reviewer for the positive and valuable feedback.
1.1) In the section on MS page 7 - line 260 "wanes" not "vanes".
We have corrected the spelling error.
1.2) Section 4.3 - the distinction between RRMS and SPMS needs to be softened. Modern thinking rejects the idea of a sudden switch from relapsing pathology to progressive pathology. This could be rewritten. I suggest citing the Lublin paper: Lublin, F.D., et al., Defining the clinical course of multiple sclerosis The 2013 revisions.Neurology, 2014. 83(3): p. 278-286.
We thank the Reviewer for this comment. We have re-written the section accordingly and included the suggested reference.
1.3) The discussion re second line treatments reflects a local view that does not have world wide applicability- suggest remove the concept of "second line" and think of some other term. ie not all countries have the same restrictions on prescribing.
We have modified the section on MS treatment strategies accordingly.
1.4) This section talks about calcium channels on endothelium and immune cells. This needs to be introduced in the section about channels on brain cells.
Following the Reviewer’s comment we have extended the introduction on p. 2 of the revised manuscript.
1.5) I cannot see the connection between clemastine and muscarinic receptors and calcium channels.
We have added some explanatory sentences regarding the connection between clemastine, muscarinic receptors and calcium on p. 10 of the revised manuscript.
Reviewer 2 Report
The review is focused on available neuroprotective drugs that is an important issue to be addressed in neuronal recovery of MS patients. This is an interesting topic that is still under debate. The manuscript is well organized but here is still some points to be discussed. It will be very useful to add the disscution part with your critical point of view about:
1) mentioned drugs positive vs negative effect;
2) choose some drugs with the most powerful positive effect that you consider to be further tested in human clinical trials in MS patients.
Best regards
Author Response
Reviewer #2
We would like to thank the Reviewer for the positive and valuable feedback.
It will be very useful to add the disscution part with your critical point of view about: 1) mentioned drugs positive vs negative effect; 2) choose some drugs with the most powerful positive effect that you consider to be further tested in human clinical trials in MS patients.
We would like to thank the Reviewer for this helpful suggestion. We have added a column to Tables 1 and 2 that lists obvious drawbacks for most of the drugs. In addition, we are now discussing the most promising candidates more extensively in the conclusion.